# Health Risk Assessment of Inorganic Mercury and Methylmercury via Rice Consumption in the Urban City of Guiyang, Southwest China

**DOI:** 10.3390/ijerph16020216

**Published:** 2019-01-14

**Authors:** Jialiang Han, Zhuo Chen, Jian Pang, Longchao Liang, Xuelu Fan, Qiuhua Li

**Affiliations:** 1College of Resources and Environmental Engineering, Guizhou University, Guiyang 550003, China; Hanjialiang627@126.com (J.H.); lianglc139@126.com (L.L.); 2School of Chemistry and Material Science, Guizhou Normal University, Guiyang 550001, China; pangjian.9132@foxmail.com (J.P.); m18335446340@163.com (X.F.); 3Key Laboratory for Information System of Mountainous Area and Protection of Ecological Environment of Guizhou Province, Guizhou Normal University, Guiyang 550001, China

**Keywords:** total mercury, methylmercury, rice, estimated daily intake, hazard quotients

## Abstract

Rice consumption is the main methylmercury (MeHg) exposure route for residents in mercury (Hg) mining areas. However, there is limited studies on mercury in commercial rice, which has high liquidity and can be directly consumed by urban residents. This study measured the total Hg (THg) and MeHg concentrations in 146 rice samples purchased from the markets in Guiyang city, southwest China, and both the inorganic Hg (IHg) and MeHg estimated daily intakes (EDIs) and hazard quotients (HQs) were calculated according to rice consumption. The THg concentrations in all rice samples (range: 0.97 to 13.10 μg·kg^−1^; mean: 3.88 μg·kg^−1^) were lower than the Chinese national standard (20 μg·kg^−1^). The average MeHg concentration in rice was 1.16 μg·kg^−1^. The total HQs (THQs) ranged from 0.0106 to 0.1048, with a mean of 0.0462, which was far lower than 1. This result suggests that there were low Hg exposure levels through consumption of commercial rice in residents of Guiyang.

## 1. Introduction

Mercury (Hg), one of the most toxic heavy metal pollutants, has been a public concern since the recognition of Minamata disease in 1956. Both inorganic and organic Hg exist in the environment. Inorganic Hg (IHg) is much less toxic than methylmercury (MeHg) to humans [1], and the absorption rate of IHg in food by the human, which is about 8% [1,2,3,4], also much less than that of MeHg (95%) [5,6]. MeHg is the most toxic organic form of organic Hg; MeHg can cross the blood-brain barrier and through placenta in humans after entering the body, causing permanent damage to the central nervous system [1,7]. The Minamata Convention, formulated by the United Nations Environment Programme, came into effect on August 16, 2017, which aimed at controlling and reducing mercury emissions globally. Fish consumption is considered to be the primary route of human MeHg exposure [6,8]; however, recent studies have reported that rice consumption is the main MeHg exposure pathway for local residents in Hg mining regions due to elevated MeHg concentrations [2,5,9,10].

As a staple food, rice constitutes 20 percent of the total food energy intake of the world’s population [11]. In Asia, the energy from rice and its by-products contributes to over 70% of residents’ daily dietary intake [12]. Studies have found that rice that grown in Hg mining areas can accumulate high concentrations of MeHg in grain, and concentrations of MeHg as high as 174 μg·kg^−1^ were recorded in brown rice, approximately 2–3 orders of magnitude higher than that in the edible parts of other crops [9,13].

Within the last decade, studies of Hg rice exposure have mainly concentrated on Hg-polluted areas, particularly at abandoned Hg mining regions in Guizhou Province, southwest China. Horvat et al. [13] first confirmed the high contamination of rice with Hg (highest concentration: 569 μg·kg^−1^ for THg; 145 μg·kg^−1^ for MeHg) in the Wanshan Hg mine in Guihou Province. Feng et al. [5] first reported a significant correlation between the estimated rice MeHg intake and hair MeHg levels in the Wanshan Hg mining region, suggesting that rice consumption was the main pathway of MeHg exposure for local populations. Recently, Li et al. [14] reported that Hg exposure via rice consumption in the Wanshan region was comparable to that via a fish diet.

With China’s rapid economic development, populations living in urbanized areas have increased quickly, and their food diets are mainly commercial rice from local markets. To date, however, few studies have been conducted on rice Hg, especially on rice MeHg in rice the markets, which is directly consumed by urban residents. Shi et al. [15] investigated THg (mean: 23 μg·kg^−1^, range: 6.3–39.3 μg·kg^−1^) and MeHg (mean: 4.7 μg·kg^−1^, range: 1.9–10.5 μg·kg^−1^) concentrations in 25 rice samples from 15 provinces in China. Qian et al. [16] thoroughly investigated THg (mean: 5.8 μg·kg^−1^, range: 0.02–31 μg·kg^−1^) concentrations in 712 samples from Chinese markets. Li et al. [17] investigated THg (mean: 10.1 μg·kg^−1^, range: 0.86–47.2 μg·kg^−1^) and MeHg (mean: 2.47 μg·kg^−1^, range: 0.13–18.2 μg·kg^−1^) concentrations in 284 rice samples from the markets of 7 provinces in China. Huang et al. [18] studied rice Hg (mean: 5 μg·kg^−1^, range: <5–88 μg·kg^−1^) from markets in Zhejiang province. Nearly all the studies existed rice Hg pollution, exceeding maximum value of 20 μg·kg^−1^ recommended by the Ministry of Public Health of China [19]. Although many studies on Hg exposure via rice consumption have been conducted in Guizhou province, which is the largest Hg producing province in China, little attention has been paid to Hg exposure via consumption of commercial rice from markets.

Therefore, Guiyang was selected as the study area, and market rice from Guiyang was collected. The purpose of this study was to assess the health risks of both inorganic Hg (IHg) and MeHg exposure in association with the consumption of commercial rice from markets in Guiyang City, the capital city of Guizhou Province. We aimed to examine and calculate (1) the concentrations of THg and MeHg in commercial rice; (2) the estimated values of the daily intakes (EDIs) of MeHg and inorganic mercury (IHg) via rice consumption; and (3) the hazard quotients (HQs) and total hazard quotients (THQs). The results will provide a better understanding of the MeHg and IHg exposure risks via consumption of commercial rice in Guiyang City, southwest China.

## 2. Methods and Materials

### 2.1. Study Area

Guiyang (106°07′ E–107°17′ E; 26°11′ N–26°55′ N) is the capital of Guizhou Province, southwest China, with an altitude of approximately 1100 meters. The annual mean temperature is 15.3 °C. The annual average relative humidity is 77%. The annual mean rainfall is 1129.5 mm. Guiyang City has a population of approximately 4.8 million, whose staple food is rice. In 2016, the output of rice in Guiyang was 17,810 tons, accounting for about 40% of the total output of cereal crops. And the per capita consumption of rice is 61.53 kg·year^−1^ (169 g·d^−1^), occupying approximately 58% of the total food consumption.

### 2.2. Sample Collection and Preparation

A total of 146 white rice samples (84 brands) were purchased from markets in Guiyang from July 22 to August 22 in 2017. The samples covered both Chinese domestic (*n* = 137) and imported rice (*n* = 9, from Cambodia, Vietnam and Thailand). According to the package information, rice samples were briefly categorized into two main subspecies—Japonica and Indica. All of the rice samples collected from the markets were transported to the laboratory, rinsed with ultrapure water, freeze-dried, ground with a grinder (IKA-A11, IKA, Staufen, Germany), and stored in plastic bags for analysis.

### 2.3. Analytical Methods

For THg analysis, approximately 0.5 g dry rice samples were weighed and were then digested at 95 °C in a water bath with a fresh mixture of HNO_3_ and H_2_SO_4_ (4:1, v/v). The THg concentrations in rice samples were determined by BrCl oxidation, SnCl_2_ reduction, purging, gold amalgamation, and cold vapour atomic fluorescence spectrometry (CVAFS) detection following Method 1631e [20].

For MeHg analysis, approximately 0.5 g of dry rice samples were weighed for digestion using 25% KOH-methanol/solvent extraction at 80 °C in a water bath. During extraction, acidification was needed after digestion of the rice sample. Then, MeHg in rice samples was extracted with dichloromethane and back-extracted into water. Finally, the concentration of MeHg was measured by gas chromatography (GC)-CVAFS according to Method 1630 [21]. IHg was obtained by the difference between THg and MeHg.

### 2.4. Quality Assurance and Quality Control

The standard reference materials (GBW(E)100359 for THg; TORT-2 for MeHg), duplicates, and method blanks were used for data quality control. The rice samples were measured in three replicates. The recoveries of THg and MeHg in the certified reference material ranged from 93% to 110% and from 90% to 115%, respectively. The detection limits for THg and MeHg were 0.013 μg·kg^−1^ and 0.003 μg·kg^−1^, respectively.

All acids, so as HNO_3_, H_2_SO_4_ and HCl (Sinopharm, Shanghai, China) used in the analysis were ultra-pure grade, and all other reagents, like KOH, were of an analytically pure grade. The CH_2_Cl_2_ (Tedia, Fairfield, OH, USA) was Chromatographic pure grade. The Hg concentrations was mensurated by the Tekran 2500 (Tekran Inc., Toronto, ON, Canada). The Brooks Rand Model III (Brooks Rand, Seattle, WA, USA) was used to measure the MeHg concentrations. The water used in the analysis was double-deionized water (DDW). All of the glassware was soaked in nitric acid for more than 24 h and washed with DDW.

### 2.5. Assessment of Human Health Risk

The estimated daily intakes (EDI) of MeHg and IHg were calculated based on the concentrations of MeHg and IHg in milled rice and rice consumed per capita according to Equation (1). The non-cancer health hazard was estimated by employing hazard quotient (HQ) Equations (2) and (3). The health risk of Hg exposure through rice consumption was estimated according to the total hazard quotient (THQ), which was calculated by Equation (4):(1)$$\mathrm{EDI}=\left(E_{F}\times E_{D}\times C\times \mathrm{IR}\times 10^{-3}\right)/\left(\mathrm{bw}\times T_{A}\right)$$
(2)$${\mathrm{HQ}}_{\mathrm{MeHg}}=\mathrm{EDI}/\mathrm{RfD}$$
(3)$${\mathrm{HQ}}_{\mathrm{IHg}}=\mathrm{EDI}/\mathrm{PTWI}$$
(4)$$\mathrm{THQ}={\mathrm{HQ}}_{\mathrm{MeHg}}+{\mathrm{HQ}}_{\mathrm{IHg}}$$

EDI is given in micrograms per kilogram of body weight per day (μg·kg^−1^·d^−1^); bw (body weight) is the average weight of the adult population. E*_F_* is the exposure frequency (365 days/year); E*_D_* is the exposure duration (70 years) and equal to the average lifetime; T_A_ is the average exposure time for non-carcinogens (365 days/year × number of exposure years, assuming 70 years in this study) [16,22]; C are the IHg and MeHg concentrations in milled rice (μg·kg^−1^); IR is the daily intake rate of rice; HQ is the hazard quotient; RfD is the reference dose for the substance; PTWI is the provisional tolerable weekly intake of inorganic mercury; and THQ is the total hazard quotient. In this study, bw is 60 kg [9]; IR is 169 g·d^−1^ [23]. RfD is 0.1 μg·kg^−1^·d^−1^ [1], indicating the maximum dose of the compound (μg·kg^−1^·d^−1^) below which there is no known hazard of health effects. The PTWI is 0.57 μg·kg^−1^·d^−1^ (equal to 4 μg·kg^−1^·week^−1^) [24,25]. HQ and THQ were used to estimate the non-cancer health hazard. When the value is >1, it indicates a potential health risk due to Hg exposure from rice consumption. If the value is <1, it is assumed to be safe according to the risk of non-carcinogenic effects. If the HQ exceeds one, there is a chance that non-carcinogenic effects may occur, with a probability which tends to increase as the value of HQ increases [26].

### 2.6. Statistical Analysis

Statistical analysis of THg and MeHg in rice was performed using Excel 2016 (Microsoft, Redmond, WA, USA), Origin 8.0 (Originlab, Northampton, MA, USA) and SPSS 22.0 (IBM, Armonk, NY, USA). Pearson correlation was employed to study the correlation coefficients. Independent-sample t tests were performed to indicate the significance of the average values of THg and MeHg in rice.

## 3. Results and Discussion

### 3.1. THg and MeHg in Rice

#### 3.1.1. Whole Market Samples

The THg concentrations in all rice samples ranged from 0.97 to 13.1 μg·kg^−1^, with a mean of 3.88 μg·kg^−1^. The THg concentrations in this study were lower than the maximum value of 20 μg·kg^−1^ recommended by the Ministry of Public Health of China [19].

Compared with the previously reported Hg values in rice from Chinese markets, the average value of 3.88 μg·kg^−1^ (*n* = 146) in rice from Guiyang market that we found was slightly lower than that reported data for THg in rice in China. In Zhoushan Island, the mean THg concentration was 9 μg·kg^−1^ (*n* = 6) [26]. Qian et al. [16] reported that the THg concentrations ranged from <0.02 to 31 μg·kg^−1^, with a mean of 5.8 μg·kg^−1^ (*n* = 712), in rice from markets in 20 provinces. In Jiangsu province, the THg concentrations ranged from 1.0 to 13 μg·kg^−1^, with a mean of 5.7 μg·kg^−1^ (*n* = 23) [27]. Li et al. [17] reported that the rice concentration from 7 provinces ranged from 0.86 to 47.2 μg·kg^−1^, with a mean of 10.1 μg·kg^−1^. Huang et al. [18] reported a mean THg concentration of 5 μg·kg^−1^ (*n* = 224) in rice from Zhejiang province. Interestingly, there is a decreasing trend of Hg in rice from markets.

For MeHg, the concentration varied from 0.069 to 2.68 μg·kg^−1^, with a mean of 1.16 μg·kg^−1^. The proportions of MeHg to THg were found to be high, reaching up to 76%, with a mean of 32.96%. Compared with the reported MeHg values in rice from nationwide markets, our results were low. Shi et al. [15] reported that the MeHg concentrations of rice from 15 provinces ranged from 1.9 to 10.5 μg·kg^−1^, with an average of 4.7 μg·kg^−1^. Li et al. [17] reported that the MeHg concentrations of rice (*n* = 284) from 7 provinces ranged from 0.13 to 18.2 μg·kg^−1^, with a mean of 2.47 μg·kg^−1^. In Changshun, for instance, the average MeHg concentration in rice was 2.5 μg·kg^−1^ (0.80–4.3μg·kg^−1^) [5], in Huaxi it was 2.9 μg·kg^−1^ (1.8–4.5 μg·kg^−1^) [28], and in Leigong it was 2.1 μg·kg^−1^ [10].

Our results for both THg and MeHg were comparable or slightly lower than results reported from markets abroad (Figure 1). Batista et al. [29] reported that the THg concentrations of 44 rice samples in Brazilian market varied from 0.3 to 13.4 μg·kg^−1^, representing most brands in the whole country. The THg and MeHg concentrations of rice samples collected from the supermarkets of Kampong city in Cambodia were in the range of 6.16–11.7 μg·kg^−1^ (a mean of 8.14 μg·kg^−1^) and 1.17–1.96 μg·kg^−1^ (a mean of 1.44 μg·kg^−1^) [30], respectively. Recently, Brombach et al. [31] reported that the THg and MeHg concentrations of 87 rice samples purchased in supermarkets in the United Kingdom, Germany and Switzerland were 3.04 ± 2.7 μg·kg^−1^ and 1.91 ± 1.07 μg·kg^−1^, respectively.

#### 3.1.2. Variations in Brands

The THg concentrations in rice of all different brands exhibited a wide range of 1.17–13 μg·kg^−1^ (Figure 2). The top ten brands with high THg concentrations were Hubeiruandingyoumi (HBRD), Taiguoxiangmi (TGXM), Hubeiruanxiangmi (HBRX), Tianfeng (TF), Huangjiadaochang (HJDC), Beiwangguo (BWG), Xingli (XL), Longyuan (LY), Tianxiaxian (TXX), and Shinong (SN), with the highest value of 13 μg·kg^−1^ recorded in HBRD. By contrast, the brands of Meinanxiang (MNX), Pingbazhuyexiangmi (PBZY), Baofa (BF), Xiangnalan (XNL), Tianxiaan (TXA), Yuyepai (YYP), Weipanwang (WPW), Daijiaxiang (DJX), Shudaoxiang (SDX) and Yongzhize (YZZ) exhibited the lowest THg concentrations, less than 2.5 μg·kg^−1^. The lowest value of 1.17 μg·kg^−1^ was observed in MNX, which was approximately one order of magnitude lower than the highest value of HBRD.

The MeHg concentrations in the samples ranged from 0.07–2.64 μg·kg^−1^. The top ten brands with high MeHg concentrations were HBDM, HBRX, TGXM, Yuchuliang (YCL), Jingxiong (JX), Jinyuanbao (JYB), TF, Mangyegu (MYG), HBRD and Tianqu (TQ). The highest value of 2.64 μg·kg^−1^ was found in the HBDM brand, and YYP exhibited the lowest value of 0.07 μg·kg^−1^, which was approximately 35 times lower than the highest value. Among the brands, 53 brands exceeded 1 μg·kg^−1^, with only 8 brands, YYP, Fengxin (FX), Mengxiang (MX), Qianlvchun (QLC), Lvhui (LH), MNX, Qianhui (QH) and Pingbazhuyexiangmi (PBZY), showing values of MeHg less than 0.5 μg·kg^−1^.

#### 3.1.3. Variations in Types

The average THg concentrations of Japonica and Indica rice were 3.96 ± 1.88 and 3.80 ± 1.90 μg·kg^−1^, respectively. The MeHg concentrations of Japonica and Indica rice were 1.10 ± 0.48 and 1.22 ± 0.57 μg·kg^−1^, respectively (Table 1). The proportion of MeHg to THg in Japonica rice (31 ± 13%) was slightly lower than that in Indica rice (35 ± 15%) (Figure 3). No significant difference of the THg (*t* test, *p* = 0.61) and MeHg (*t* test, *p* = 0.17) concentrations between Japonica rice and Indica rice were observed in the present study.

The THg concentrations in Japonica rice were similar to those in Japonica rice (*n* = 24) (mean: 3.46 μg·kg^−1^; range: 0.73–10 μg·kg^−1^), as reported by Brombach et al. [31]; however, the MeHg concentrations of Japonica rice (1.63 μg·kg^−1^) were slightly higher than those found in the present study. Previous studies reported that different rice types may influence the MeHg concentrations in rice [27,33,34].

### 3.2. Risk Assessment via Rice Consumption

The EDIs of IHg via rice consumption in the present study ranged from 0.0012 to 0.0326 μg·kg^−1^·d^−1^, with a mean of 0.0076 μg·kg^−1^·d^−1^. Compared to the Wanshan mining area, these values were 1–2 orders of magnitude lower than those reported by Li et al. [2], in which the EDI_IHg_ values through rice consumption in two sites were 0.3191 and 0.1632 μg·kg^−1^·d^−1^, respectively.

The EDI_IHg_ values determine through consuming Japonica rice were 0.0027–0.0326 μg·kg^−1^·d^−1^ (0.008 μg·kg^−1^·d^−1^), which were in the same range as the EDI_IHg_ values of 0.0012–0.0317 μg·kg^−1^·d^−1^, with a mean of 0.0073 μg·kg^−1^·d^−1^ via Indica rice consumption (Table 1).

The EDI_IHg_ values of different brands ranged widely, from 0.0017 to 0.0317 μg·kg^−1^·d^−1^ (Figure 4). The top ten brands by EDI_IHg_ value were HBRD, BWG, XL, TXX, HJDC, TF, TGXM, QLC, LY and QL, the peak value of 0.0317 g·kg^−1^·d^−1^ recorded in HBRD. By contrast, BF, TXA, MNX, WPW, SDX, DJX, XGYJ, AMN, YZZ and PBZY exhibited the lowest EDI_IHg_ values, and the lowest value of 0.0017 g·kg^−1^·d^−1^ was found in in BF, which was approximately 18 times lower than the highest value in HBRD.

The HQs of IHg exposure via rice consumption in the present study ranged from 0.0021 to 0.0570, with a mean value of 0.0134, which were one order of magnitude lower than those reported by Li et al. [2]. Considering the different brands, the HQ_IHg_ values ranged from 0.0029 to 0.0554, and the HQ_IHg_ of Japonica and Indica rice were 0.0047–0.0570 (mean: 0.0141) and 0.0021–0.0554 (mean: 0.0127), respectively. The HQs of IHg exposure via rice consumption in this study were less than 1, which indicated there was no health risk of IHg exposure via rice consumption for residents in Guiyang.

The EDIs of MeHg exposure through rice consumption in the present study were in the range of 0.0002–0.0076 μg·kg^−1^·d^−1^. The top ten highest and lowest EDI_MeHg_ values from different brands are showed in Figure 4. These EDI_MeHg_ values in different brands ranged from 0.0002 to 0.0074 μg·kg^−1^·d^−1^, with a mean of 0.0033 μg·kg^−1^·d^−1^. The highest EDI_MeHg_ value of 0.0074 μg·kg^−1^·d^−1^ was found for HBDM, which was approximately 35 times higher than that of the lowest EDI_MeHg_ value for YYP. Considering the types of rice, the EDI_MeHg_ values via consumption of Japonica and Indica rice ranged from 0.0002 to 0.0076 μg·kg^−1^·d^−1^, with a mean of 0.0031 μg·kg^−1^·d^−1^, and 0.0002–0.0075 μg·kg^−1^·d^−1^, with a mean of 0.0034 μg·kg^−1^·d^−1^, respectively.

Based on EDI_MeHg_ and RfD, the calculated HQ_MeHg_ values in the present study were between 0.0020 and 0.0756. Approximately 90% of the HQ_MeHg_ values from the different brands were distributed between 0.0100 and 0.0600 (Table 2). Table 1 showed that the mean HQ_MeHg_ values of Japonica rice and Indica rice were 0.0311 (0.0020–0.0756) and 0.0345 (0.0023–0.0750), respectively. In comparison, the EDIs and HQs of MeHg from Japonica rice and Indica rice were within the same range, exhibiting no significant differences.

Compared to the HQ_MeHg_ values reported in the literature, our data were comparable or lower than those reported from non-contaminated areas as well as markets. The HQ_MeHg_ values found in the present study were lower than the HQ_MeHg_ values found in the background site (a mean of 0.13 ± 0.052), as reported by Rothenberg et al. [33]. Li et al. [17] reported the HQ values of MeHg through rice consumption, ranging from 0.04 to 0.08, in 7 provinces in China. The highest HQ_MeHg_ value of 0.0756 in this study was far below the highest HQ_MeHg_ value of 0.39 via rice consumption of rice purchased from European supermarkets [31]. Moreover, the HQ_MeHg_ values for residents in Guiyang were also lower than the HQ_MeHg_ values reported for residents of Hg mine areas in China [9,35].

In the present study, the THQ values ranged from 0.0106 to 0.1048, with a mean of 0.0462, which was far below 1. This result might suggest low Hg exposure through the consumption of rice purchased from markets in Guiyang for the residents. Our data suggest that residents should regularly change the brands they consume.

## 4. Conclusions

The study illustrated the THg and MeHg concentrations in the commercial rice samples from Guiyang. Results showed that THg concentrations were lower than the national tolerance limit of 20 μg·kg^−1^ in rice, suggesting a safe level of Hg. There was no significant difference in the THg and MeHg concentrations of the different types of rice. Low THQs (<1), based on the EDIs values, indicate the low Hg exposure of rice consumers in Guiyang. Though the generally low concentrations of THg and MeHg in rice from Guiyang markets, we suggest that the residents should consider different brands to avoid relatively high Hg exposure.

## Figures and Tables

**Figure 1 ijerph-16-00216-f001:**
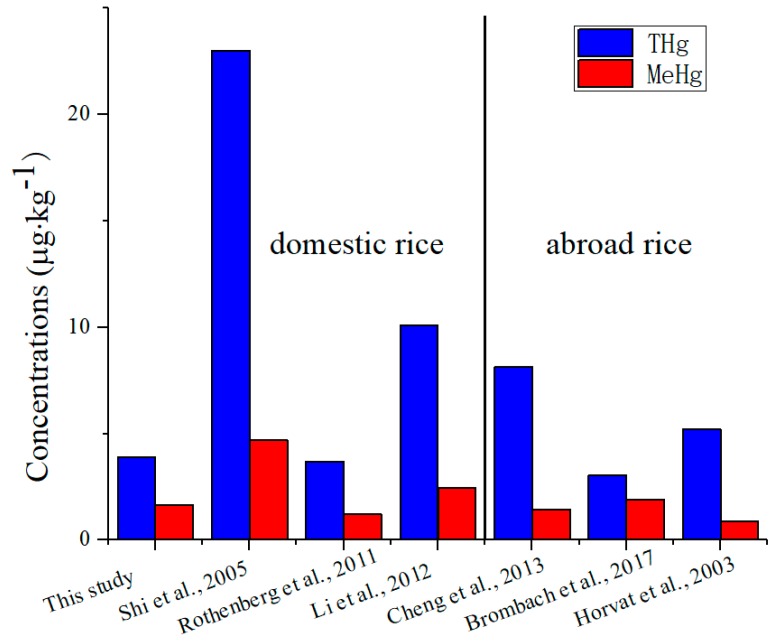
The THg and MeHg concentrations from rice in different study. Note: The samples collected by Horvat et al. were from Italy [13]; by Shi et al. from 15 provinces of China [15]; by Li et al. from 7 provinces in southern China [17]; by Cheng et al. from Cambodia [30]; by Brombach et al. from the Europe [31] and by Rothenberg et al. from Hubei province in China [32].

**Figure 2 ijerph-16-00216-f002:**
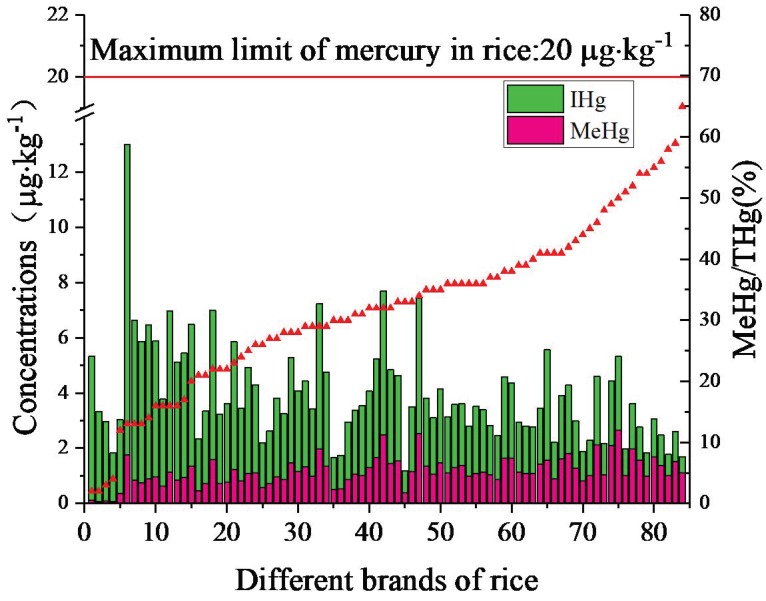
The THg and MeHg concentrations, %MeHg (of THg) of different brands rice.

**Figure 3 ijerph-16-00216-f003:**
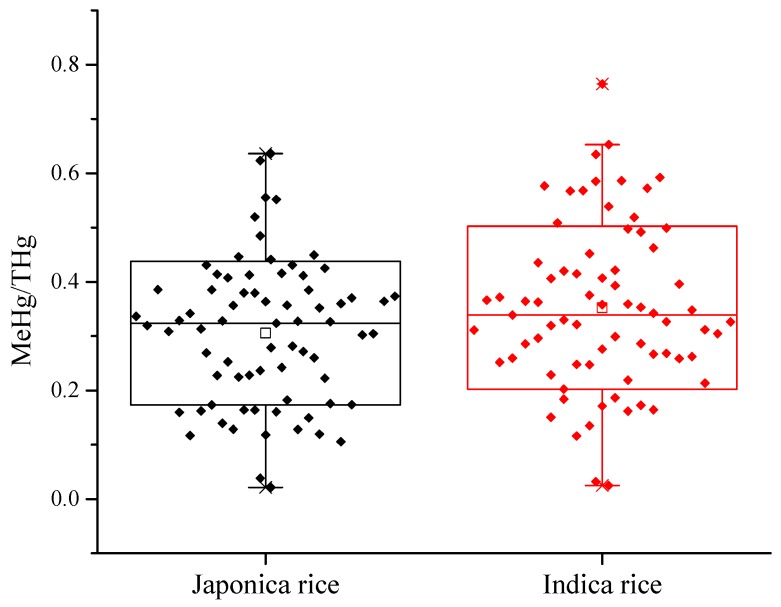
The proportion of the MeHg to THg in Japonica and Indica rice.

**Figure 4 ijerph-16-00216-f004:**
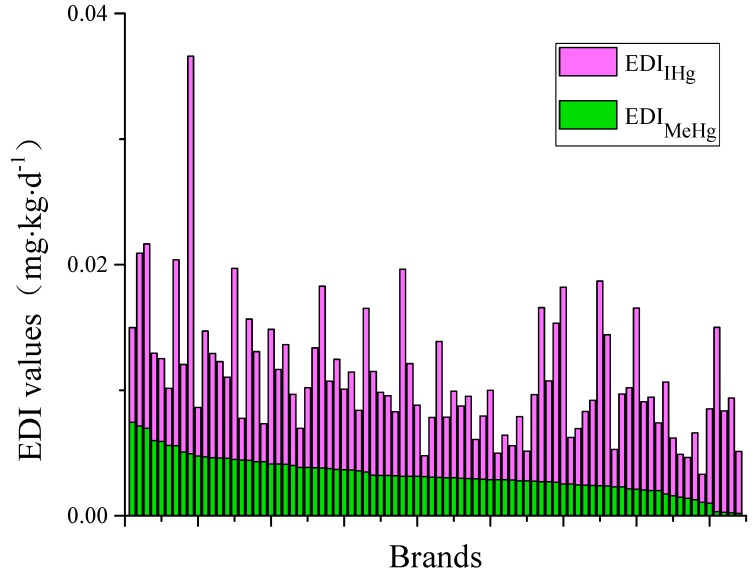
The EDI (estimated daily intakes) value from the different brands of rice.

**Table 1 ijerph-16-00216-t001:** The mean values of THg, MeHg, MeHg/THg, IHg, EDI and HQ of MeHg and IHg, and THQ in Japonica and Indica rice.

Types	THg (μg·kg^−1^)	MeHg (μg·kg^−1^)	MeHg/THg (%)	IHg (μg·kg^−1^)	EDI_MeHg_ (μg·kg^−1^·d^−1^)	HQ_MeHg_	EDI_IHg_ (μg·kg^−1^·d^−1^)	HQ_IHg_	THQ
Japonica rice (*n* = 73)	3.96 (1.71–13.10)	1.10 (0.07–2.68)	31 (2–64)	2.86 (0.96–11.56)	0.0031 (0.0002–0.0076)	0.0311 (0.0020–0.0756)	0.0080 (0.0027–0.0326)	0.0141 (0.0047–0.0570)	0.0452 (0.0106–0.1004)
Indica rice (*n* = 73)	3.80 (0.97–13.00)	1.22 (0.08–2.66)	35 (2–76)	2.57 (0.42–11.25)	0.0034 (0.0002–0.0075)	0.0345 (0.0023–0.0750)	0.0073 (0.0012–0.0317)	0.0127 (0.0021–0.0554)	0.0472 (0.0147–0.1048)

**Table 2 ijerph-16-00216-t002:** The mean values of THg, MeHg, MeHg/THg, IHg, HQ of MeHg and IHg, and THQ indifferent brands of rice.

Brands	THg (μg·kg^−1^)	MeHg (μg·kg^−1^)	MeHg/THg (%)	IHg (μg·kg^−1^)	HQ_MeHg_	HQ_IHg_	THQ
AMN (*n* = 1)	2.47 ± 0.02	1.37 ± 0.22	56%	1.10	0.0386	0.0054	0.0440
BDH (*n* = 2)	5.22 ± 0.70	1.67 ± 0.17	32%	3.55	0.0469	0.0175	0.0644
BF (*n* = 1)	1.70 ± 0.28	1.11 ± 0.08	65%	0.59	0.0312	0.0029	0.0341
BFYR (*n* = 1)	2.28 ± 0.05	1.02 ± 0.05	45%	1.26	0.0287	0.0062	0.0349
BNM (*n* = 1)	3.91 ± 0.13	1.62 ± 0.14	41%	2.29	0.0456	0.0113	0.0570
BWG (*n* = 1)	6.97 ± 0.14	1.12 ± 0.17	16%	5.85	0.0315	0.0289	0.0604
CSH (*n* = 3)	3.40 ± 0.46	1.14 ± 0.63	36%	2.26	0.0321	0.0111	0.0433
CW (*n* = 1)	3.49 ± 0.09	1.14 ± 0.18	33%	2.35	0.0322	0.0116	0.0438
CZB (*n* = 1)	2.82 ± 0.11	1.03 ± 0.12	37%	1.79	0.0291	0.0088	0.0379
DBDM (*n* = 1)	2.63 ± 0.09	0.71 ± 0.09	27%	1.91	0.0201	0.0094	0.0295
DJX (*n* = 1)	1.87 ± 0.18	0.82 ± 0.05	44%	1.06	0.0230	0.0052	0.0282
FLM (*n* = 10)	2.78 ± 0.57	1.09 ± 0.30	40%	1.69	0.0308	0.0083	0.0391
FX (*n* = 1)	3.33 ± 0.03	0.083 ± 0.005	2%	3.25	0.0023	0.0160	0.0184
GF (*n* = 1)	4.64 ± 0.12	1.53 ± 0.07	33%	3.11	0.0431	0.0153	0.0585
GFXR (*n* = 1)	3.44 ± 0.04	1.43 ± 0.08	41%	2.01	0.0402	0.0099	0.0501
GTX (*n* = 1)	3.38 ± 0.11	1.05 ± 0.09	31%	2.33	0.0297	0.0115	0.0412
HBDM (*n* = 1)	5.32 ± 0.14	2.64 ± 0.11	50%	2.67	0.0745	0.0132	0.0877
HBRD (*n* = 1)	13.00 ± 2.38	1.75 ± 0.07	13%	11.25	0.0494	0.0554	0.1048
HBRX (*n* = 1)	7.42 ± 0.19	2.54 ± 0.10	34%	4.88	0.0714	0.0241	0.0955
HDXA (*n* = 3)	4.84 ± 1.22	1.46 ± 0.40	32%	3.38	0.0410	0.0167	0.0577
HF (*n* = 1)	3.13 ± 0.02	1.12 ± 0.18	36%	2.01	0.0314	0.0099	0.0413
HJDC (*n* = 1)	7.00 ± 0.07	1.60 ± 0.10	23%	5.40	0.0449	0.0266	0.0716
HJDY (*n* = 1)	4.08 ± 0.04	1.15 ± 0.07	28%	2.93	0.0324	0.0145	0.0469
HL (*n* = 2)	2.76 ± 2.53	1.57 ± 1.54	54%	1.19	0.0443	0.0058	0.0502
HLX (*n* = 2)	3.81 ± 1.09	0.96 ± 0.24	27%	2.85	0.0271	0.0140	0.0412
HY (*n* = 2)	3.26 ± 1.13	0.86 ± 0.07	28%	2.40	0.0242	0.0118	0.0360
HZH (*n* = 3)	2.46 ± 0.43	0.87 ± 0.28	37%	1.59	0.0246	0.0078	0.0324
JCYP (*n* = 1)	4.36 ± 0.05	1.64 ± 0.09	38%	2.72	0.0461	0.0134	0.0595
JJ-1 (*n* = 4)	3.10 ± 0.80	1.06 ± 0.08	35%	2.04	0.0297	0.0101	0.0398
JJ-2 (*n* = 2)	4.42 ± 0.55	1.31 ± 0.61	29%	3.11	0.0370	0.0153	0.0523
JLY (*n* = 9)	3.54 ± 1.70	1.02 ± 0.47	31%	2.52	0.0288	0.0124	0.0412
JX (*n* = 2)	4.44 ± 1.06	2.10 ± 0.01	49%	2.34	0.0593	0.0115	0.0708
JYB (*n* = 3)	3.60 ± 0.84	1.99 ± 1.02	52%	1.62	0.0559	0.0080	0.0639
KK (*n* = 2)	5.27 ± 0.76	1.47 ± 0.01	28%	3.80	0.0413	0.0188	0.0601
KS (*n* = 1)	4.30 ± 0.11	1.12 ± 0.10	26%	3.18	0.0315	0.0157	0.0472
LDT (*n* = 1)	3.58 ± 0.03	1.30 ± 0.11	36%	2.28	0.0367	0.0112	0.0480
LF (*n* = 1)	3.36 ± 0.13	0.72 ± 0.08	21%	2.64	0.0202	0.0130	0.0332
LFHT (*n* = 1)	2.94 ± 0.06	1.13 ± 0.18	39%	1.81	0.0319	0.0089	0.0408
LFY (*n* = 3)	5.86 ± 2.16	1.24 ± 0.14	23%	4.63	0.0348	0.0228	0.0576
LH (*n* = 1)	3.02 ± 0.10	0.36 ± 0.04	12%	2.66	0.0102	0.0131	0.0233
LY (*n* = 2)	6.49 ± 1.56	1.36 ± 0.68	20%	5.14	0.0382	0.0253	0.0635
LZ (*n* = 2)	4.59 ± 1.23	1.64 ± 0.41	38%	2.95	0.0462	0.0145	0.0607
MG (*n* = 3)	3.62 ± 0.77	1.37 ± 0.62	36%	2.25	0.0386	0.0111	0.0497
MNX (*n* = 1)	1.17 ± 0.29	0.38 ± 0.02	33%	0.79	0.0108	0.0039	0.0147
MPS (*n* = 2)	4.93 ± 2.17	1.08 ± 0.03	25%	3.84	0.0306	0.0190	0.0495
MSJL (*n* = 1)	5.12 ± 0.20	0.84 ± 0.10	16%	4.28	0.0237	0.0211	0.0448
MX (*n* = 1)	2.97 ± 0.03	0.09 ± 0.01	3%	2.87	0.0027	0.0142	0.0168
MYG (*n* = 1)	4.28 ± 0.04	1.80 ± 0.10	42%	2.48	0.0508	0.0122	0.0630
NJM (*n* = 2)	3.23 ± 0.16	0.73 ± 0.28	22%	2.50	0.0205	0.0123	0.0328
PBZY (*n* = 1)	1.65 ± 0.09	0.49 ± 0.03	30%	1.16	0.0139	0.0057	0.0196
QFDY (*n* = 1)	3.45 ± 0.07	0.81 ± 0.13	24%	2.63	0.0229	0.0130	0.0359
QH (*n* = 2)	2.34 ± 0.53	0.46 ± 0.20	21%	1.88	0.0129	0.0093	0.0222
QL (*n* = 1)	5.87 ± 0.06	0.75 ± 0.05	13%	5.12	0.0212	0.0252	0.0464
QLC (*n* = 1)	5.32 ± 0.06	0.11 ± 0.01	2%	5.21	0.0031	0.0257	0.0288
QSC (*n* = 1)	4.07 ± 0.74	1.30 ± 0.05	32%	2.77	0.0366	0.0136	0.0502
SDX (*n* = 1)	1.98 ± 0.12	1.01 ± 0.06	51%	0.97	0.0284	0.0048	0.0332
SN (*n* = 1)	5.88 ± 0.12	0.96 ± 0.15	16%	4.92	0.0272	0.0243	0.0514
SXY (*n* = 1)	4.14 ± 0.11	1.46 ± 0.07	35%	2.68	0.0412	0.0132	0.0543
SY (*n* = 5)	5.56 ± 4.43	1.57 ± 0.28	41%	3.99	0.0442	0.0197	0.0639
TF (*n* = 2)	7.24 ± 3.07	1.98 ± 0.23	29%	5.25	0.0559	0.0259	0.0818
TGXM (*n* = 1)	7.69 ± 0.20	2.47 ± 0.11	32%	5.22	0.0696	0.0257	0.0953
TH (*n* = 1)	2.21 ± 0.07	0.90 ± 0.08	41%	1.31	0.0253	0.0065	0.0318
TQ (*n* = 1)	3.06 ± 0.08	1.69 ± 0.08	55%	1.37	0.0476	0.0068	0.0543
TXA (*n* = 1)	1.77 ± 0.10	1.02 ± 0.05	58%	0.75	0.0287	0.0037	0.0324
TXX (*n* = 1)	6.46 ± 0.06	0.90 ± 0.06	14%	5.56	0.0254	0.0274	0.0528
WC (*n* = 2)	5.44 ± 0.63	0.95 ± 0.21	17%	4.50	0.0267	0.0222	0.0488
WF (*n* = 4)	3.62 ± 1.08	0.76 ± 0.45	22%	2.86	0.0214	0.0141	0.0355
WH (*n* = 4)	2.95 ± 0.47	0.87 ± 0.15	30%	2.08	0.0245	0.0102	0.0347
WMCX (*n* = 1)	3.78 ± 0.13	0.61 ± 0.07	16%	3.17	0.0173	0.0156	0.0329
WPW (*n* = 1)	1.83 ± 0.30	0.99 ± 0.07	54%	0.84	0.0278	0.0042	0.0320
XBB (*n* = 1)	2.98 ± 0.07	1.27 ± 0.11	43%	1.72	0.0357	0.0085	0.0442
XGYJ (*n* = 1)	2.60 ± 0.08	1.53 ± 0.11	59%	1.08	0.0430	0.0053	0.0483
XL (*n* = 1)	6.63 ± 0.06	0.85 ± 0.05	13%	5.78	0.0240	0.0285	0.0525
XMY (*n* = 4)	2.80 ± 0.35	0.99 ± 0.09	36%	1.82	0.0278	0.0090	0.0368
XNL (*n* = 1)	1.74 ± 0.43	0.53 ± 0.03	30%	1.21	0.0149	0.0060	0.0209
XRJ (*n* = 4)	3.52 ± 1.89	1.08 ± 0.44	36%	2.45	0.0303	0.0121	0.0423
XXBX (*n* = 1)	2.20 ± 0.22	0.57 ± 0.03	26%	1.63	0.0160	0.0080	0.0240
XZXM (*n* = 1)	3.43 ± 0.11	0.98 ± 0.08	29%	2.45	0.0276	0.0121	0.0397
YCL (*n* = 1)	4.59 ± 0.18	2.12 ± 0.11	46%	2.47	0.0598	0.0122	0.0720
YD (*n* = 1)	3.80 ± 0.08	1.34 ± 0.21	35%	2.46	0.0377	0.0121	0.0498
YYP (*n* = 1)	1.82 ± 0.07	0.07 ± 0.01	4%	1.75	0.0020	0.0086	0.0106
YZXM (*n* = 1)	4.74 ± 0.12	1.36 ± 0.06	29%	3.39	0.0382	0.0167	0.0549
YZZ (*n* = 1)	2.16 ± 0.02	1.05 ± 0.17	48%	1.11	0.0295	0.0055	0.0349
ZJB (*n* =1)	2.79 ± 0.07	1.08 ± 0.05	39%	1.71	0.0303	0.0085	0.0388

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
