# Peer review of "Health Risk Assessment of Inorganic Mercury and Methylmercury via Rice Consumption in the Urban City of Guiyang, Southwest China"

_ijerph, 2019, doi:10.3390/ijerph16020216_

Round 1
Reviewer 1 Report (Reviewer 1 is Reviewer 3 for the original submission)
Thank you for considering my comments.
Please change in line:
123 "PTWI is the provisional tolerable weekly intake of rice" to "PTWI is the provisional tolerable weekly intake of inorganic mercury".
There is no need to duplicate the names Japonica rice and Indica rice in Figure 4. The names are already under the X axis.
Author Response
Reviewer #1’: Thank you for considering my comments.
Response: Thanks for your constructive suggestions.
Please change in line:
123 "PTWI is the provisional tolerable weekly intake of rice" to "PTWI is the provisional tolerable weekly intake of inorganic mercury".
Response: We thank the reviewer for this constructive suggestion. And we have already revised the definition of the PTWI according to your suggestion. Please see line 131-132.
There is no need to duplicate the names Japonica rice and Indica rice in Figure 4. The names are already under the X axis.
Response: Thanks reviewer for this suggestion. Based on your constructive suggestion, we have already revised the Figure 4. Please see the Figure 3.
Reviewer 2 Report
General comments
This paper reports the concentrations of total mercury and methylmercury in rice samples from markets in Guiyang city. The authors studied the influence of brands on these concentrations and discuss about the human risk assessment through rice consumption based on estimated daily intake.
The results obtained are interesting and this article could be used as a reference for others studies carried out in contaminated areas of China. However, the major shortcoming in the paper is the presentation of the data. Some parts should be more detailed to improve the readability of the paper.
In general, I am not convinced by the English language and I think that the manuscript needs revisions.
In conclusion, I have several major and minor comments aimed to improve the quality and readability of the paper.
Specific comments
Introduction
Lines 53-62: the authors said that few studies have been conducted on Hg-rice from local markets. However, several studies are cited. What are the results in these studies? Do these studies show a problem according to the concentrations of Hg in market-rice? Please provide better the importance to continue to study rice and Hg from markets.
Methods and Materials
Line 82: “imported rice”: from where?
Line 87: 0.2-0.5 g: the range is wide. Why not a precise mass? It could have a bias in the results of concentrations.
Line 116: in the calculation of EDI, where are the parameters: EF (exposure frequency), ED (exposure duration) and AT (averaging time)? I guess you consider EFxED/AT=1 but it is necessary to give this information and explain your choice.
Lines 117 and 118: Why did you use two different reference dose: RfD and PTWI. Explain your choice.
Line 129: and for carcinogenic effects?
Results and Discussion
Lines 140-148 and lines 149-156: because it is very difficult for a non-Chinese scientist to locate the different provinces, maybe it will be interesting to a have a sort of cartography, e.g. a map with the different zones (provinces) and put the values on the map (color representation). Proposal: bring together this map with a part of Fig. 1.
Line 158: Why the reference “Batista et al. [29]” is not represented in the Fig. 1?
Fig. 1: abroad rice: where exactly. Proposal: put provinces or countries instead of references.
Lines 167 and 182: the study of influence of brands and types on mercury concentrations in rice should be presented in the objectives at the end of the introduction.
Big question: what is the difference between “brands”, “types” and “varieties”?
Line 168: why the range 1.17-13 µg/kg is different of the range line 137. It is on the same population (i.e. all samples)?
Lines 169-173: what is the meaning of HBRD, TGXM…. It is interesting to compare the results between the different brands but for a non-Chinese scientist, it is very difficult to understand. What are the main differences between these brands?
Line 183: Japonica and Indica rice: what is it? Not defined in the Methods and Materials.
Lines 192-193: but not in the present study. How do you explain your results?
Table 2: this table is very long. What is the interest to present all the results for each brand of rice? Proposal: put only a statistical distribution of your data (min, mean, max and percentile values).
Section 3.2.2.: why change the data presentation for MeHg? I suggest to bring together sections 3.2.1 and 3.2.2, and then Table 1 and Fig 5 (in one only table), and to finish Table 2 and Fig 6 (in one only table).
Lines 30-31, page 12: The results showed that the non-carcinogenic risk is very low through consumption of rice from markets. So, why do you recommend changing regularly the brands?
Conclusions
Lines 39-41, page 12: as the previous comment, why this further investigation?
Author Response
Reviewer #2: Specific comments
Introduction
Lines 53-62: the authors said that few studies have been conducted on Hg-rice from local markets. However, several studies are cited. What are the results in these studies? Do these studies show a problem according to the concentrations of Hg in market-rice? Please provide better the importance to continue to study rice and Hg from markets.
Response: Thank you for your advice, we have added the values of them, and also emphasize the importance of study in Guiyang. Please see line 57-67.
Methods and Materials
Line 82: “imported rice”: from where?
Response: the rice samples we collected from Cambodia, Vietnam and Thailand. We added the information in Line 89.
Line 87: 0.2-0.5 g: the range is wide. Why not a precise mass? It could have a bias in the results of concentrations.
Response: Thanks for your suggestion. The mass of rice we weight in the experiment is 0.5 g. And we have already corrected it in the manuscript. Please see line 94.
Line 116: in the calculation of EDI, where are the parameters: EF (exposure frequency), ED (exposure duration) and AT (averaging time)? I guess you consider EFxED/AT=1 but it is necessary to give this information and explain your choice.
Response: We also considered the EF (365 days/year), ED (70 year) and AT (365 days/year number of exposure years, assuming 70 years in this study) (Zheng et al., 2007; Qian et al.,2010). After calculating the EF*ED/AT=1 can be obtained, then we simplify the equation according to Li et al. (2012). And we also provide the explanation in Line 122 and127-130.
Lines 117 and 118: Why did you use two different reference dose: RfD and PTWI. Explain your choice.
Response: Thank you for the question. Actually, RfD is for MeHg, while PTWI is for IHg in our study. And both of them were obtained on the fish consumption, because fish contains many protein and important nutrients (such as omega-3 fatty acids, and various vitamins and minerals), which will substantially reduce the toxicity of mercury after fish consumption, however, rice has no these nutrients like fish. Thus, the stricter limit RfD was chosen in this study.
Line 129: and for carcinogenic effects?
Response: Thank you for your advice. Since Hg is a non-carcinogenic heavy metal, and the HQ and THQ represent non-carcinogenic effects, thus we only calculate non-carcinogenic effects in our study. Carcinogenic effects were used for carcinogenic heavy metals, such as Cd, Cr, Pb.
Results and Discussion
Lines 140-148 and lines 149-156: because it is very difficult for a non-Chinese scientist to locate the different provinces, maybe it will be interesting to a have a sort of cartography, e.g. a map with the different zones (provinces) and put the values on the map (color representation). Proposal: bring together this map with a part of Fig. 1.
Response: Thank you for your advice. Actually, our objectives are to investigate the market rice in Guiyang, and evaluate the associated risk via rice consumption for Guiyang residents. Meanwhile, our sampling scale is too small to represent rice Hg levels of one province, since there exist thousands of rice brands in Chinese markets. Also, there exist a few imported rice samples, thus we prefer to provide some information about its producing areas. Hope you will be satisfied with our answer.
Line 158: Why the reference “Batista et al. [29]” is not represented in the Fig. 1?
Response: Thanks for your reminding. But the Batista et al. (2012) only gave the range of the concentrations of THg and did not give the mean of the THg. And that also had no the values of the MeHg in rice. So we did not represent the “Batista et al.” in the Fig. 1. Otherwise, we have already added the Horvat et al. (2003) into the Fig. 1.
Fig. 1: abroad rice: where exactly. Proposal: put provinces or countries instead of references.
Response: Thanks for your valuable advice. Actually, we tried to add the province names of these studies in China, while the study areas are usually covering 7-15 provinces. Thus, there is no enough space to add in the figure, and we added some information in the caption of Fig.1.
Lines 167 and 182: the study of influence of brands and types on mercury concentrations in rice should be presented in the objectives at the end of the introduction.
Response: Thank you for your reminding. Since our goal is to investigate the commercial rice in the markets, and different rice types in the markets were usually packaged in different brands. Thus, we discuss the samples according to the briefly rice types, and brands. Hope you will satisfied with our answer.
Big question: what is the difference between “brands”, “types” and “varieties”?
Response: In our study the types and varieties have the same means, which refer to the Japonica rice and Indica rice. And the rice samples were produced from the different manufactures. We changed “varieties” into “types”.
Line 168: why the range 1.17-13 µg/kg is different of the range line 137. It is on the same population (i.e. all samples)?
Response: Thank you for your advice. Actually. Line 137 is results of the total Hg of all the rice samples. Meanwhile, line 168 is the results of the total Hg of the different brands of rice. Considering that different brands of rice contain the different amounts of samples, we used the mean values of the total Hg to represent the same one brand of rice.
Lines 169-173: what is the meaning of HBRD, TGXM…. It is interesting to compare the results between the different brands but for a non-Chinese scientist, it is very difficult to understand. What are the main differences between these brands?
Response: Thank you for your question, the meaning of HBRD, TGXM …represent the ID of different brands. Since rice samples were collected with different brands, we want to show the rice Hg level in Guiyang, and give the idea that commercial rice in Guiyang is safe to local residents.
Line 183: Japonica and Indica rice: what is it? Not defined in the Methods and Materials.
Response: Thank you for your question, Japonica and Indica rice are the two major subspecies of cultivated rice. We added in this explanation in Lines 89-90.
Lines 192-193: but not in the present study. How do you explain your results?
Response: Thank you for your comment, we actually want to cite this reason to explain why the methylmercury of Japonica rice is a little higher than Indica in this study. Also, we try to avoid the conclusion expression since we have no data in our study. Thus, we choose to delete the sentence “Therefore, different types of rice may have different MeHg concentrations”.
Table 2: this table is very long. What is the interest to present all the results for each brand of rice? Proposal: put only a statistical distribution of your data (min, mean, max and percentile values).
Response: Thank you for your advice, we changed according to your comments.
Section 3.2.2.: why change the data presentation for MeHg? I suggest to bring together sections 3.2.1 and 3.2.2, and then Table 1 and Fig 5 (in one only table), and to finish Table 2 and Fig 6 (in one only table).
Response: Thanks for your question, we changed the data because the former reviewer advised us to remove the consideration of adsorption rates of IHg (7%) and MeHg (95%). Actually, when evaluating Hg exposure for only one source, it is no need to consider the absorption. Absorption rates were only used when calculating different exposure source. Hope our explanation is enough to answer your question. We put all of them together according to your valuable comments. We combined 3.2.1 and 3.2.2 together according to you point. Also, we really appreciate your pointing out, some of figures and tables are repeated. We combined the Table 2 and Figure 6, however, for Table 1 and Figure 3 are expressing same contents, rather Figure 5, then we put Figure 3 and Table 1 together. Hope this is a good way to satisfy your comments.
Lines 30-31, page 12: The results showed that the non-carcinogenic risk is very low through consumption of rice from markets. So, why do you recommend changing regularly the brands?
Response: Thank you for your good question. Since from the data the rice THg and MeHg can reach up to 13.1 μg·kg-1 and 2.68 μg·kg-1, which represent the highest level and exposure. We suggest residents to change brands to avoid highest rice THg and MeHg.
Conclusions
Lines 39-41, page 12: as the previous comment, why this further investigation?
Response: Thank you for your constructions’ comment; we deleted the expression “The future study should pay more attention on the … to THg were observed”.
References:
1. Zheng, N.; Wang, Q.; Zhang, X.; Zheng, D.; Zhang, Z.; Zhang, S. Population health risk due to dietary intake of heavy metals in the industrial area of Huludao city, China. Sci. Total Environ. 2007, 387, 96-104.
2. Qian, Y.Z.; Chen, C.; Zhang, Q.; Li, Y.; Chen, Z.J.; Li, M. Concentrations of cadmium, lead, mercury and arsenic in Chinese market milled rice and associated population health risk. Food Control, 2010,21, 1757-1763.
3. Li, P.; Feng, X.B.; Yuan, X.B.; Chan, H.M.; Qiu, G.L.; Sun, G.X.; Zhu, Y.G. Rice consumption contributes to low level methylmercury exposure in southern China. Environ. Int. 2012, 49, 18-23.
Round 2
Reviewer 2 Report
I can see that the authors made the required revisions to the manuscript, or explained why they did not accept a few remarks. In general, the manuscript has been restructured to improve its quality and readability. Moreover, the different points underlined by the reviewers have been explained and detailed. I don't know if the English language was checked. This manuscript has been greatly improved.
This manuscript is a resubmission of an earlier submission. The following is a list of the peer review reports and author responses from that submission.
Round 1
Reviewer 1 Report
See file attached

Author Response
Reviewer #1’: The manuscript is interesting to publish in the journal with some revisions. The results of the research are useful and interesting from scientific point of view. Title – well describe the article. Abstract - well describe the article. Keywords - - well describe. Introduction – well described.
Response: Thanks reviewer for comments.
Experimental – well describe all information but some points need clarification: Par 2.4 Quality assurance and quality control – The author indicates the concentration of THg and MeHg in the certified material. Separate the sentence in the line 99- 102 in other paragraph “Material, reagents and instrumentation” and describe the materials, reagents and instrumentation used for the work, insert also the purity of reagent, solvent and the manufacturer.
Response: We thank the reviewer for this constructive suggestion. Then according to the suggestion of reviewer, we have been separated the sentence in the line 99- 102 in another paragraph. And we have been added the purity of reagent, solvent and the manufacturer. See lines 102-107.
Results and Discussion - well describe the results obtained but some points need clarification: line 202-204 pg 7 – explain the acronims of the brands eventually in a separate table because do not understand the meaning in the table 2 and in the text.
Response: Thanks reviewer for comment. Generally, since all the brands of rice in Guiyang are safe to the consumers, we think it is no need to reveal the name of them. Furthermore, if we reveal the name of them, it may affect the businesses of the manufacturers. So we think it is not necessary to list the specific name of each brand.
References – well describe.
Response: Thanks reviewer for comments.
Conclusion – It is interesting to know the reason of the lower value of THg and MeHG in the rice compared to the other area reported in the paper.
Response: Actually, we do think it is quite nice results. This may be due to the strict supervision on markets and producing areas.
Table and figure - well described.
Response: Thanks reviewer for comments.
Reviewer 2 Report
The introduction of the paper can be improved making mention of Minamata convention and informing total amount of potential inhabitants exposed in Guiyang area.
In 2.5 Assessment of human health risk: please to inform absorption rate of Hg (¿do you use the same for methyl and inorganic mercury?, usually methyl mercury is more absorbed than inorganic one. I suggest to complete the analysis with child scenario of exposure in order to protect most vulnerable people. And finally is very interesting for foreign lectors to know the current daily intake rate of rice for adult and child in China. Please inform it.
Author Response
Reviewer #2: The introduction of the paper can be improved making mention of Minamata convention and informing total amount of potential inhabitants exposed in Guiyang area.
Response: We thank the reviewer for this constructive suggestion. And we have added the description of the Minamata convention at the line 32-34. And in 2.1 Study area, we have already mentioned that the population of Guiyang is 4.8 million.
In 2.5 Assessment of human health risk: please to inform absorption rate of Hg (¿do you use the same for methyl and inorganic mercury?, usually methyl mercury is more absorbed than inorganic one.
Response: In 2.5 Assessment of human health risk: We mentioned that absorption efficiencies of Hg species in diet by human body are considered as approximately 8% for inorganic mercury and for 95% Methylmercury. Please see line 122-123.
I suggest to complete the analysis with child scenario of exposure in order to protect most vulnerable people. And finally is very interesting for foreign lectors to know the current daily intake rate of rice for adult and child in China. Please inform it.
Response: We thank the reviewer for this constructive suggestion. For the reason, we did not conduct a questionnaire survey and there was no specific daily intake rate of rice for children. And the daily intake rate of other cities is not suit for Guiyang. Thus, we did not analyze the child scenario of exposure. The daily intake of rice is 115.3-229.5 g·d-1 for 2-14 years old children, 207.5-266.1 g·d-1 for 14-18 years old children and 224.9-272.6 g·d-1 for adult according to the Nutrition and Health Status of Chinese residents (2008).
Reference:
Gao, J. S. Investigation report on nutrition and health status of Chinese Residents Ten: 2002 nutrition and health data set [M]. Beijing: People Health Press, 2008, 47-156.
Reviewer 3 Report
The manuscript presents monitoring data on Hg concentrations in rice available on the local market in Guiyang, southwest China.
Firstly, authors described the analytical methodology which seems to be well performed and appropriate quality controls were applied. The only question I have to the authors is whether they measured Hg concentration in solution obtained after washing the rice before powdering.
The second part of the manuscript is devoted to health risk assessment. This part needs fundamental revision. Hazard quotients are calculated as a ratio between EDI (estimated daily intake) and PTWI or RfD (please explain the rationale for selection of RfD not the PTWI for organic mercury). Authors incorrectly considered absorption rate (Ai) in the equation (1) since PTWI and RfD are established based on the oral doses in the pivotal studies. Ai should be deleted from this equation.
Additionally, in the case of inorganic mercury (IHg), EDI should be multiplied by 7 (days) in the equation (3) since PTWI represents 7-day exposure.
Above mentioned suggestions will significantly change the conclusions.
Other comments:
Fig. 6. The HQ values should have no unit on the Y axis.
Author Response
Reviewer #3: Firstly, authors described the analytical methodology which seems to be well performed and appropriate quality controls were applied. The only question I have to the authors is whether they measured Hg concentration in solution obtained after washing the rice before powdering.
Response: Firstly, thanks reviewer for comments. We did not measure the Hg concentration in the water of washing the rice. Since residents always do the washing before cooking, then we think it is not necessary to measure Hg washing.
The second part of the manuscript is devoted to health risk assessment. This part needs fundamental revision. Hazard quotients are calculated as a ratio between EDI (estimated daily intake) and PTWI or RfD (please explain the rationale for selection of RfD not the PTWI for organic mercury). Authors incorrectly considered absorption rate (Ai) in the equation (1) since PTWI and RfD are established based on the oral doses in the pivotal studies. Ai should be deleted from this equation.
Response: Thank you for your suggestions. For the reason we choose RfD, since the PTWI and RfD were both obtained on the fish consumption, and fish contains many protein and important nutrients (such as omega-3 fatty acids, and various vitamins and minerals), which will substantially reduce the toxicity of mercury after fish consumption. However, rice has no these nutrients like fish. This means mercury in rice is more toxic than it in fish. Thus, we prefer to choose RfD to calculate the hazard quotients.
For the adsorption rate, according to Guidance for Identifying Populations at Rish from Mercury Exposure (WHO, 2008) , since the exposure formula was obtained on the fish consumption, and >95% methylmercury will be absorbed into the gastrointestinal tract, then the adsorption rate was roughly considered as 100%. In most of studies, the adsorption rate of methylmercury was considered as 95%, and 8% for inorganic mercury (Rahola et al., 1973; Clarkson et al., 2006; Feng et al., 2008: Li et al., 2015). Also, the reviewer 2 emphasized the importance of different adsorption rate of methylmercury and inorganic mercury. Thus, we prefer to use Ai as a part of equation (1).
Additionally, in the case of inorganic mercury (IHg), EDI should be multiplied by 7 (days) in the equation (3) since PTWI represents 7-day exposure.
Above mentioned suggestions will significantly change the conclusions.
Response: We thank reviewer the suggestion. About the hazard quotient of the inorganic mercury, actually we used 0.57 μg·kg-1·d-1 (equal to 4 μg·kg-1·week-1) as PTWI to hazard quotient in the equation (3). And we clarified it in line 125.
Other comments:
Fig. 6. The HQ values should have no unit on the Y axis.
Response: We thank reviewer the constructive suggestion, and we have already corrected it, please see Fig.6 .
Reference:
1. Rahola, T.; Hattula, T.; Korolainen, A.; Miettinen, J.K. Elimination of free and protein-bound ionic mercury (20Hg2+) in man. Ann. Clin. Res. 1973, 5, 214-219.
2. Clarkson T.W.; Magos, L. The toxicology of mercury and its chemical compounds. Crit. Rev. Toxicol. 2006, 36, 609-662.
3. WHO, “Guidance for Identifying Populations at Risk from Mercury Exposure,” United Nations Environment Programme, WHO Department of Food Safety, Zoonoses and Foodborne Diseases, Geneva, Switzerland, 2008. http://www.unep.org/hazardoussubstances/Mercury/MercuryPublications/GuidanceTrainingmaterialToolkits/GuidanceforIdentifyingPopulationsatRisk/tabid/3616/language/enUS/Default.aspx.
4. Feng, X.; Li, P.; Qiu, G.L.; Wang, S.F.; Li, G.H.; Shang, L.H.; Meng, B.; Jiang, H.M.; Bai, W.Y.; Li, Z.G.; Fu, X.W. Human exposure to methylmercury through rice intake in mercury mining areas, Guizhou province, China. Environ. Sci. Technol. 2008, 42, 326-332.
5. Li, P.; Du, B.Y.; Chan, H.M.; Feng, X.B. Human inorganic mercury exposure, renal effects and possible pathways in Wanshan mercury mining area, China. Environ. Res. 2015, 140, 198-204.
Round 2
Reviewer 3 Report
Dear Authors,
thank you for your answers, however I cannot agree with your argumentation to include Ai in the equation (1). HQ is calculated as a ratio between EDI and RfD (organic) or PTWI (inorganic). Both RfD and PTWI already assume different absorption rato of mercury compounds - simply: RfD and PTWi represent an INTAKE not the internal dose. EDI represent INTAKE not the internal dose. When you consider Ai in the equation you will get not EDI but internal dose but in this case you cannot compare this value to RfD or PTWI.
According to your argumentation the risk related to inorganic mercury is underestimated roughly 12 times (assuming 8% absorption rate).